# Understanding GDPR Non-Compliance in Privacy Policies of Alexa Skills in European Marketplaces

## ABSTRACT

Amazon Alexa is one of the largest Voice Personal Assistant (VPA) platforms and it allows third-party developers to publish their voice apps, named skills, to the Alexa skill store. To satisfy the needs of European users, Amazon Alexa established multiple skill marketplaces in Europe and allows developers to publish skills in their native languages, such as German, French, Italian, and Spanish. Skills targeting users in European countries are required to comply with GDPR (General Data Protection Regulation), which imposes strict obligations on data collection and processing. Skills that involve data collection should provide a privacy policy to disclose the data practice to users and meet GDPR requirements.

In this work, we analyze privacy policies of skills in European marketplaces, focusing on whether skills' privacy policies and data collection behaviors comply with GDPR. We collect a large-scale European skill dataset that includes skills in all European marketplaces with privacy policies. To classify whether a sentence in a privacy policy provides GDPR information, we gather a labeled dataset consisting of skills' privacy policy sentences and train a BERT model for classification. Then we analyze the GDPR compliance of European skills. Using a dynamic testing tool based on ChatGPT, we check whether skills' privacy policies comply with GDPR and are consistent with the actual data collection behaviors. Surprisingly, we find that 67% of privacy policies fail to comply with GDPR and don't provide necessary GDPR-related information. For 1,187 skills with data collection behaviors, we find that 603 skills (50.8%) don't provide a complete privacy policy and 1,128 skills (95%) have GDPR non-compliance issues in their privacy policies. Meanwhile, we find that the GDPR has a positive influence on European privacy policies when compared to non-European marketplaces, such as the United States, Mexico and Brazil.

## 1 INTRODUCTION

Nowadays, Voice Personal Assistants (VPA), such as Amazon Alexa and Google Assistant, are popular in people's daily lives and significantly change users' lifestyles [11]. Amazon Alexa is one of the largest VPA platforms, allowing third-party developers to publish their voice apps (named skills) to the Alexa skill store. This approach significantly increases the number of Alexa skills available and enhances Alexa's functionality, enabling actions such as playing music or ordering food from restaurants. To satisfy the growing demand of users in European countries, Amazon Alexa built several European marketplaces in addition to the United States and allowed developers to publish skills in their local languages, *e.g.*, German, French, Italian, and Spanish. These non-English skills are expected to provide better services to local users in their native language.

Alexa skills might collect users' sensitive personal data for specific functions, such as searching for nearby restaurants by using the user's location and designing customized services for users using their names. In such cases, Alexa requires skills to provide a privacy policy to disclose the data practices to users [2]. A privacy policy document should inform users about what information is collected, how the information is used and what information is being shared [9]. However, existing works [19, 30, 32, 43] reveal that privacy policies provided by third-party developers were often poorly written. Meanwhile, VPA platforms don't evaluate the quality of privacy policy content during the certification process [32]. Such results indicate the overlook of user privacy by both VPA platforms and developers.

In the US marketplace, data collection behaviors in skills are restricted by different legal and lawful regulations, *e.g.*, CalOPPA (California Online Privacy Protection Act) [4], CCPA (California Consumer Privacy Act) [3], COPPA (Children's Online Privacy Protection Act) [5], and HIPAA (Health Insurance Portability and Accountability Act) [10]. Similarly, Europe has implemented the General Data Protection Regulation (GDPR) [7] in 2018 to enhance individual's control and rights over their personal data. Non-compliance with GDPR in privacy policies could result in substantial fines. For example, in 2019, Google was fined €50 million by the French government because of its failure to provide complete privacy policies that comply with the GDPR [8].

While most existing works analyze GDPR compliance for diverse software applications [34–36], none of them focus on the Alexa skills regarding GDPR compliance in privacy policies. In this work, we analyze the privacy policies of Alexa skills in European marketplaces and check whether they comply with the GDPR. The diversity of languages in European privacy policies poses a challenge to our analysis. Also, there is no existing dataset of the Alexa skill privacy policies or model for classifying a skill's privacy policy sentences in the context of GDPR. In addition, the skill behaviors may differ from what they claim in privacy policies. Thus, it is essential to test the actual behaviors of skills and compare them with privacy policies to check privacy policy consistency. In this work, our primary objective is to assess the extent to which privacy policies associated with Alexa skills available in European marketplaces align with the GDPR in order to promote the provision of responsible and GDPR-compliant privacy notices on the Web.

In summary, we have the following contributions:

- We gather an Alexa skill privacy policy dataset about the GDPR and train a BERT model for predicting if a sentence within privacy policies is about GDPR. We translate non-English privacy policies into English and use the BERT model to identify GDPR-related sentences.
- We collect 23,927 privacy policies of all European skills and conduct a large-scale analysis of their GDPR non-compliance. We find that most of the privacy policies (67%) in European marketplaces don't comply with GDPR. We also find that the GDPR has a positive influence on European privacy policies when compared to non-European marketplaces.
- We implement a dynamic testing tool based on ChatGPT to test skills in European marketplaces. Then we detect data

collection behaviors in skills and check skills' privacy policy compliance. We find 169 skills lacking a privacy policy, 434 skills providing an incomplete one and 22 skills with a deceptive one. We also find that 1,128 skills collecting data have GDPR non-compliance issues.

- We have shared our dataset, model, and results with the community to facilitate future research. The details of our work are available at https://github.com/Alexa-skills-GDPR/Alexa-skills-GDPR.

## 2 BACKGROUND

### 2.1 Alexa Skill Marketplaces

Amazon Alexa skill store is one of the most prominent voice app platforms and it has over 100,000 skills worldwide. Such a large skill number benefits from Alexa's allowance for third-party developers to publish their own skills to the skill store. After establishing the United States (US) marketplace in 2016 and observing the rapid increase in Alexa skill numbers, Amazon expanded marketplaces to other countries in 2017, including Australia (AU), Canada (CA), India (IN), and the United Kingdom (GB), etc. Amazon encouraged developers to migrate their English skills to these new marketplaces or develop new skills specifically for them. After the initial success, Amazon expanded by establishing non-English marketplaces in Germany (DE), France (FR), Spain (ES), and Italy (IT), targeting European users. Due to the difficulty in translating skills to a new language and the need of re-developing a new skill, the skill number in these marketplaces is much less than that in English-speaking marketplaces. However, the skills in native languages can provide better service to local users. In this work, we focus on the European marketplaces, including GB, DE, ES, IT and FR, and conduct a large-scale analysis of their privacy policies. In addition, we use the US, ME (Mexico), and BR (Brazil) marketplaces for comparison and discuss the potential influence of GDPR on European marketplaces.

### 2.2 Privacy Policy and GDPR Categories

| Category | Article | Label |
|----------|---------|-------|
| 1 | 13.1 | Collect Personal Information |
| 2 | 13.2 (a) | Data Retention Period |
| 3 | 13.1 (c) | Data Processing Purposes |
| 4 | 13.1 (a)(b) | Contact Details |
| 5 | 13.2 (b) | Right to Access |
| 6 | 13.2 (b) | Right to Rectify or Erase |
| 7 | 13.2 (b) | Right to Restrict of Processing |
| 8 | 13.2 (b) | Right to Object to Processing |
| 9 | 13.2 (b) | Right to Data Portability |
| 10 | 13.2 (d) | Right to Lodge a Complaint |

Table 1: GDPR categories in Article 13

A privacy policy is a legal document that discloses how a party gathers, uses, discloses, and manages a customer's data. To protect users' data and privacy better, Alexa requires all skills that collect user data to provide a privacy policy. The privacy policy link would be displayed on the skill's listing webpage, accompanied by the skill name, developer, description, etc. The Alexa platform may prevent a skill from publication if it doesn't follow Alexa's privacy requirements. In addition, the privacy policy is important for users as it is the primary channel for users to learn about what and how

data will be used before users enable and invoke a skill. However, developers may not provide the required privacy policy or provide an incomplete privacy policy inadvertently or intentionally [32, 43].

The GDPR is a regulation enforced since 2018 in Europe and European Economic Area (EEA) about data collection and privacy. GDPR is one of the strictest data protection laws and it has 11 chapters with 99 articles about various data requirements, *e.g.*, lawfulness, fairness, data ministration, and accountability. In particular, article 13 of GDPR discusses the "information to be provided where personal data are collected from the data subject". Since such information should be included in the privacy policy, we focus on analyzing whether a skill's privacy policy contains such information if it collects user data. The GDPR categories mentioned in Article 13 are listed in Table 1. Specifically, GDPR requires that "When personal data are collected from the data subject, the controller shall provide the data subject with **all of the following information**", so when category 1 is involved in a skill, information regarding all the other categories should also be provided.

## 3 OUR APPROACH AND DATA COLLECTION

**Methodology Overview:** Figure 1 shows the overview of our approach. First, we gather a labeled skill privacy policy dataset about GDPR (details in Section 4.1). Then we use it to train a BERT model to classify privacy policy sentences into GDPR categories. Since our model is trained for English sentence classification, we translate privacy policies from European languages into English and use our BERT model to identify GDPR-related information in a privacy policy. Second, based on the BERT model prediction, we measure whether a skill's privacy policy in European marketplaces complies with GDPR (GDPR Compliance). For a privacy policy, if it claims to collect user data, we check whether it provides the necessary information defined in the GDPR categories listed in Table 1. If not, it fails to comply with the GDPR requirements. Lastly, we perform a dynamic testing of skills in European marketplaces. Then, we check whether skills' privacy policies are consistent with skill data collection behaviors (Privacy Policy Consistency). If a skill collects user data through the voice channel or permission requests, it should provide a privacy policy and mention the data collection. If not, it has a potential privacy policy issue such as lacking a privacy policy or having an incomplete/deceptive privacy policy. In addition, skills with data collection behaviors may also have GDPR non-compliance in their privacy policies.

**Data Collection:** We collected our Alexa skill dataset in March 2022, including the US, GB, DE, ES, IT, and FR marketplaces. In addition, we added the ME and BR marketplaces for comparison. We developed a web crawler that automatically visited each category and every skill listed in each marketplace during the data collection. For each skill, we collected the skill ID, skill name, developer, utterances, description, privacy policy link, and permissions. The number of skills and skills with a privacy policy in each marketplace are shown in Table 2. Interestingly, the two earliest marketplaces (US and UK) have the lowest percentage of skills with a privacy policy. After obtaining the privacy policy links of skills, we retrieved the content of these privacy policies. Since some websites use JavaScript and will dynamically load contents after visiting, we used the Selenium Webdriver [13] to visit each website in a

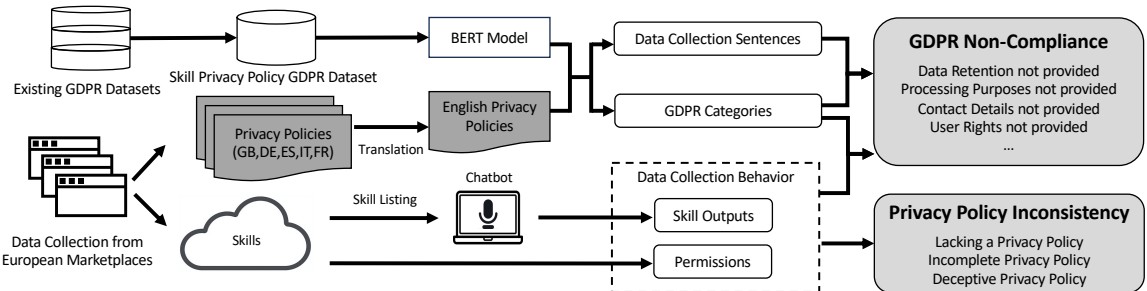

**Figure 1: Methodology Overview**

browser so that all the content could display correctly. For privacy policies in HTML format, we used the "trafilatura" [14] library to extract the text and store the content in a text file. For PDF files, we used the "pdf2text" [12] library to transfer the file to a text file. At last, we split each privacy policy into separate sentences and removed blank lines. For the US marketplace, we only downloaded the privacy policies of US skills that are also published in other non-English marketplaces for comparison. We implemented the data collection and all our analyses in Python code.

| Marketplaces | # of Skills | # of Skills with PP | Percentage | Year Created |
|---|---|---|---|---|
| US | 76,306 | 22,241 | **29%** | 2016 |
| UK | 40,812 | 9,484 | **23%** | 2016 |
| Germany | 10,626 | 3,976 | 37% | 2018 |
| Spain | 5,802 | 2,032 | 35% | 2019 |
| Italy | 5,181 | 2,586 | 50% | 2019 |
| France | 3,156 | 1,673 | 53% | 2018 |
| Mexico | 2,778 | 1,280 | 46% | 2021 |
| Brazil | 1,940 | 1,053 | 54% | 2021 |

**Table 2: Number of skills and skills with a privacy policy in each marketplace.**

## 4 CLASSIFICATION MODEL

### 4.1 Skill Privacy Policy Dataset on GDPR

Due to the high requirement for understanding the GDPR articles and the necessity of proficiency to label a GDPR dataset (researchers in [35] recruited 22 knowledgeable annotators for data annotation), we chose not to label a new dataset by ourselves but to use existing GDPR datasets provided by three existing works [34–36], which were labeled by domain experts and had higher reliability. We found that some companies may use the same privacy policy link for all the products, *e.g.*, the company website, Android app, browser extension, as well as Alexa skill. Therefore, we searched for the privacy policies that are used by skills and labeled in existing datasets. For example, 8 Amazon official skills and the Amazon app on Google Play use the same privacy policy link, which is Amazon's official privacy notice. We first downloaded the three existing privacy policy datasets and merged them after removing duplicates. After that, we searched for each skill's privacy policy link in three existing datasets, which provide both the privacy policy links and labeled sentences within privacy policies. Note that the three datasets are in English, so we only searched for the skills within the US marketplace. Finally, we obtained 113 privacy policies of Alexa skills, and these privacy policy documents contain 2,586 labeled sentences.

### 4.2 GDPR Model Training

After getting the skill privacy policy dataset, we used the three datasets as training data and tested different models on the skill privacy policy dataset to evaluate their performance, with an objective to choose the optimal model in our analysis. Note that we excluded any data from the three datasets if it was present in the skill privacy policy dataset. We trained and compared different models for privacy policy sentence classification, *e.g.*, DNN [37], BiLSTM [23] and BERT [18]. As a result, the BERT (Bidirectional Encoder Representations from Transformers) model outperforms others, which has been proved in previous works [34, 35].

We trained a BERT-Base model with 12 layers, 12 attention heads, and 768 hidden vectors. During the model training, we used the Adam algorithm for optimization and searched the learning rate within [1e-5, 3e-5, 5e-5, 1e-4,3e-4,5e-4,1e-3], batch size within [4,8,16,32,64,128], and epochs within [10, 20, 50] to find the best optimal parameters. With the learning rate 3e-5 and the batch size 32 within 10 epochs, our model obtained a weighted F1 score of 80.3 on the validation data and 80.1 on our skill privacy policy dataset. We also manually checked 100 predicted sentences and 81 were correct, which shows a reasonable performance. We used the trained BERT model for sentence classification and predicted all privacy policy sentences regarding GDPR categories . It costs 4 hours to predict all the 23,927 privacy policies in European marketplaces.

### 4.3 Privacy Policy Translation

**Challenges.** Since it is hard to recruit experts to label datasets for each European language separately, we decided to translate privacy policies from different European languages to English since we only have the model for English sentence classification. To find a stable and effective translation method, we tried and compared several State-of-the-art APIs, including Google Translate, Argos Translate, and Reverso Translation. However, two challenges still exist. First, we need a quantitative method to evaluate the translation performance based on the translated text. Second, after the translation, we need to obtain paired English and non-English sentences with the same meaning to evaluate translation performance. However, manually translating non-English privacy policy sentences into English for each language is non-trivial.

To address the first challenge, we evaluated the translation performance based on our trained GDPR model and we checked how much the translation methods influence the performance. This is because the translated sentences serve as input for model classification. Thus, we focused on assessing the translation's overall impact on the model rather than other metrics such as sentence

similarity. For two sentences with the same meaning, one is in English and the other is translated from another language, if our trained model has the same predictions on the two sentences, we considered the translation process has minimal influence on the model performance.

For the second challenge, we solved it by searching for the privacy policy pairs in different marketplaces. There exist skills that were published in several marketplaces with different languages. Developers may translate the English privacy policy directly into another language. For such cases, the sentences in English and non-English privacy policies are semantically aligned and have the same meaning. For example, the skill "WOOX Security" provides six versions of privacy policy links corresponding to different languages. Moreover, after checking these privacy policies, we found that almost all the content, titles and headings are the same except for the language. Figure 2 in the Appendix shows its privacy policies in English and Spanish. For such skills, we can obtain the privacy policy pairs in different languages with the same content and use their sentences to evaluate the translation performance.

**Privacy Policy Alignment.** To obtain the corresponding English sentences in non-English privacy policies, we proposed a method to align and get the privacy policy pairs. We first collected all skills published in the US marketplace (English) with a privacy policy link and all skills in European marketplaces in the local language (German, French, Spanish, and Italian). Then, we downloaded the privacy policies of these skills in different languages. After that, we aligned the content of each skill's privacy policy in the non-English version with the one in English and this process includes four steps.

First, we translated a non-English privacy policy into English and split the two privacy policies into sentences. Second, we calculated the similarity of each sentence in the two privacy policies. If the summed similarity of three continuous sentences is over a threshold (0.8 for each sentence and 2.4 in sum), we considered the sentence block and three sentences aligned. We didn't compare similarities based on individual sentence since some simple sentences might have several similar sentences in another privacy policy. Third, after parts of sentences have been aligned, we checked whether the sentences between aligned sentences are same. For example, suppose we have aligned the 1st and 2nd titles in two privacy policies. If the number of sentences under the 1st title are same in the two privacy policies, we can determine that all sentences between the two titles are aligned. Fourth, we checked how many sentences in the two privacy policies are aligned. If the percentage of aligned sentences exceeds a threshold (80% in our work), we considered the two privacy policies aligned and we used the aligned sentences in our evaluation. As a result, we obtained 99 skills with aligned privacy policy pairs in European languages for evaluating the translation performance.

**Translation Performance.** We evaluated three translation methods on the 99 skills with aligned privacy policies. We first used our trained BERT model to predict the English privacy policy sentences as the baseline. After that, we checked the model outputs of translated privacy policies and compared the differences. Table 3 shows the performance (weighted F1-score) of three translation methods in four different languages. Overall, we found that Google Translate performs better than the others and the translation process doesn't influence the model performance much. We also noticed that the

translation performances for different languages are similar, motivating us to apply Google Translate to all the European languages in this work. We use Google Translate to translate all European privacy policies into English.

| Marketplace | Language | Google | Argos | Reverso |
|---|---|---|---|---|
| DE | German | 92.8 | 90.5 | 85.2 |
| ES | Spanish | 93.7 | 91.3 | 88.7 |
| IT | Italian | 93.7 | 92.6 | 89.3 |
| FR | French | 94.5 | 92.2 | 86.4 |

**Table 3: Performance (weighted F1-score) of different translation methods.**

## 5 GDPR NON-COMPLIANCE ANALYSIS

After translating privacy policies into English and predicting the sentences using our trained BERT model, we checked their GDPR compliance. Since GDPR targets personal data collection, we first checked whether a privacy policy claims that it collects user data by searching for sentences labeled "Collect Personal Information" (Category 1). Unlike other platforms, *e.g.*, Android apps and browser extensions, voice assistants can only collect certain types of personal data, and some data, such as IP addresses or cookies, can't be collected in voice apps. Therefore, we added a filter for sentences predicted as "Collect Personal Information" and only considered the sentences containing certain data types, including name, email, address, birthday, age, and location, etc. The complete list of data types is obtained from the SkillDetective [43]. If no sentence mentions collecting user data, the privacy policy doesn't have GDPR non-compliance issues. If any sentence is labeled as "Collect Personal Information", we checked whether the other GDPR categories appear in the predicted results, as mentioned in Section 2.2. If any category is missing, the privacy policy doesn't comply with the GDPR requirements and a GDPR violation exists. For each marketplace, we calculated the number of skills claiming data collection and average violations per skill. We found that *72% of European skills claim they collect data but 67% of skills have GDPR violations in privacy policies.*

### 5.1 GDPR Non-Compliance per Category

Table 4 shows the results of GDPR violations. We first calculated the percentage of violations in each GDPR category. When comparing different categories in GDPR, we noticed that "Data Processing Purpose" (4.12%) and "Contact Details" (5.22%) have the lowest percentage of violations. This is possible because most privacy policies would mention what they use the data for after claiming their data collection, such as "We collect your name for ...". For the "Contact Details" category, in more cases, developers/companies would leave their names in privacy policies. On the contrary, the categories about the rights of the data subjects, *i.e.*, the right to "Object to Processing" (47.03%), "Data Portability" (53.37%), and "Lodge a Complaint" (54.64%), have more violations. Since these categories are unique in GDPR, developers may not notice or know them and thus do not provide the corresponding information in the privacy policies. For example, although the skill "World Traveler" mentions the user's right to access to personal information, other rights are not addressed in the privacy policy. The "Wedding Table Finder" skill (shown in Figure 3 in the Appendix) provides a good

| GDPR Category | European Marketplaces | | | | | Comparison | | | |
|---|---|---|---|---|---|---|---|---|---|
| | GB | DE | ES | IT | FR | US | ME | BR | Average |
| Data Retention Period | 40.88% | 21.55% | 38.18% | 41.66% | 37.58% | 31.70% | 36.48% | 43.69% | 35.97% |
| Data Processing Purposes | 5.85% | 2.83% | 4.03% | 3.93% | 3.96% | 4.97% | 3.71% | 7.30% | 4.12% |
| Contact Details | 3.61% | 5.37% | 5.15% | 6.00% | 6.00% | 6.08% | 4.44% | 9.57% | 5.22% |
| Right to Access | 34.61% | 51.45% | 40.27% | 35.86% | 42.34% | 42.05% | 43.91% | 45.07% | 40.91% |
| Right to Rectify or Erase | 38.80% | 22.18% | 36.44% | 45.71% | 33.00% | 27.10% | 30.59% | 41.32% | 35.23% |
| Right to Restrict of Processing | 29.81% | 21.26% | 32.36% | 32.13% | 35.11% | 36.13% | 34.54% | 37.38% | 30.13% |
| Right to Object to Processing | 48.69% | 28.86% | 54.59% | 53.81% | 49.20% | 45.43% | 50.36% | 60.65% | 47.03% |
| Right to Data Portability | 54.79% | 34.72% | 55.50% | 57.98% | 53.83% | 49.45% | 55.93% | 60.65% | 53.37% |
| Right to Lodge a Complaint | 57.06% | 37.50% | 62.64% | 57.39% | 58.59% | 55.15% | 62.79% | 70.71% | 54.64% |
| # of Skill with Data Collection | 5,898 | 2,965 | 1,460 | 1,838 | 1,207 | 1,291 | 898 | 794 | |
| # of Skill with Violations | 5,536 | 2,602 | 1,407 | 1,750 | 1,126 | 1,213 | 851 | 775 | |
| Percentage of Skill with Violations | 93.86% | 87.76% | 96.37% | 95.21% | 93.29% | 93.96% | 94.77% | 97.61% | |
| Average Violations per Skill | 3.82 | 3.03 | 4.04 | 4.07 | 3.94 | 3.66 | 3.95 | 4.55 | 3.73 |

**Table 4: GDPR violations of all marketplaces. The category and marketplaces with the most violations are highlighted.**

example of privacy policy adhering to GDPR requirements. For each GDPR category, there exists a section to disclose the data usage. Specifically, the section "What are your data protection rights?" lists and discusses the details of each user right individually, which correspond to the GDPR categories. Obviously, this privacy policy is written for the skill being published in Europe.

## 5.2 GDPR Non-Compliance per Marketplace

Table 4 also shows the number of skills with data collection, the number of skills with GDPR violations and the average number of violations per skill in each marketplace. Surprisingly, 72% of skills claim they collect data in privacy policies in all European marketplaces. Among these skills, 93% of them have GDPR violations and each skill has 3.73 violations on average. We compared marketplaces in different areas to understand the impact of GDPR.

First, we compared the US and GB marketplaces to understand whether developers would change their privacy policies in response to GDPR without the need of changing languages. For the two English marketplaces, we found the average GDPR violations per skill doesn't change much (3.66 vs. 3.82). This is because most skills published in the GB marketplace are migrated from the US marketplace and developers may not modify the privacy policies at all. Such results indicate that most developers don't read the GDPR requirements and make corresponding changes when publishing skills to European marketplaces. This poses a risk to both developers and the Amazon Alexa platform. Given that GDPR is a lawful regulation, the platform should inform developers about GDPR requirements. Failing to do that could expose Amazon to penalties from European authorities, similar to Google's fine in 2019 [8].

Second, we compared European marketplaces (GB, DE, ES, IT, FR), which all should follow the GDPR requirements. Compared to the GB marketplace (average of 3.82 violations per skill), we found more violations in non-English marketplaces, except the DE marketplace (average of 3.03 per skill). The ES (4.04) and IT (4.07) marketplaces have the highest average violations per skill. This is possibly because they directly use more privacy policies written in English (58% and 48%), which are not written for European marketplaces and don't consider the GDPR. The DE marketplace has the lowest percentage of privacy policies in English because more privacy policies were specifically written for the European marketplaces and they follow the GDPR better.

Lastly, we compared the European marketplaces against the ME and BR marketplaces, which are not in English and not subject to

GDPR. Our objective is to find out whether GDPR helps to improve the quality of non-English privacy policies. We found that the ME (3.95) and BR (4.55) marketplaces have more violations per skill than the European marketplaces (3.73 per skill on average), indicating that *the GDPR does have a positive influence on the privacy policy quality in European marketplaces.*

## 5.3 Skills Published in Multiple Marketplaces

To find out more possible reasons why each marketplace has different violation numbers, we checked details about the skills published in more than one marketplace and how privacy policies change between two or more marketplaces.

By comparing the US marketplace with the GB marketplace, we found that 6,373 skills were published in the two marketplaces and 6,094 (96%) skills have the same privacy policy in the two marketplaces. Among them, 1,358 skills claim they collect data in privacy policies but have GDPR violations in both marketplaces. 279 skills change their privacy policies in the GB marketplace. Among them, only 14 skills improve their privacy policies to comply with GDPR in the GB marketplace. We even found 7 skills worsened: their privacy policies were compliant with GDPR in the US marketplace but didn't comply in the GB marketplace.

When comparing the skills published in the GB and other European marketplaces, we observed that 53% of the skills use the same privacy policy in the two marketplaces (the GB and another marketplace) and 70% of these skills have GDPR violations. For the skills with different privacy policy links in two European marketplaces, only 8% of skills enhance their privacy policy in non-English, while 10% diminish the quality regarding GDPR compliance. Figure 4 in the Appendix shows a skill named "T'nB Smart" and it provides two privacy policies in English and French separately. The English privacy policy ends with Section 4 and in the FR marketplace, it adds two additional sections about user's rights and GDPR. We consider such a skill as an enhancement in the privacy policy.

We also compared the 583 skills that were published in the US and all the 5 European marketplaces. Most skills don't claim they collect data in privacy policies. For other skills that claim to collect data, 94 skills have violations in all marketplaces and 10 skills don't have violations in all marketplaces. We are more interested in the other 78 skills with different privacy policies in each marketplace and how they behave differently. *Overall, the European marketplaces have fewer skills with GDPR violations than the US marketplace.* On the contrary, the ME and BR marketplaces have more skills with

violations and perform much worse than the US and European marketplaces. The percentages of skill with GDPR violations are 49%, 52%, and 59% for the European, US and non-European marketplaces (ME and BR), respectively. These results indicate that *the GDPR has a positive influence on privacy policies when ignoring the language factor* (the GB marketplace outperforms the US). Furthermore, *the GDPR has a significantly positive influence on privacy policies in European marketplaces.*

We also investigated the changes of GDPR non-compliance in European marketplaces based on a recent dataset collected in 2023. We observed that the average GDPR violation number per skill decreased in almost all European marketplaces. The details of these findings can be found in Appendix A.

# 6 PRIVACY POLICY INCONSISTENCY ANALYSIS

In Section 5, we checked the sentences in a privacy policy to determine whether it collects user data and complies with GDPR. However, the privacy policy might not reflect the actual data collection behaviors in skills and previous works [19, 32, 43] have revealed that a large number of privacy policies in the skill store were of low quality. Developers might hide their data collection behaviors and not mention that in the privacy policies. In this section, we first checked two types of data collection behaviors in skills: asking for user data through the voice channel outputs and requesting data collection permissions. Then we checked the GDPR non-compliance and privacy policy inconsistency by comparing the privacy policies against the actual behaviors of skills.

## 6.1 Dynamic Testing of Skills Using ChatGPT

To find the data collection behaviors in skill outputs, we needed to dynamically test skills since the skill code is hosted on the cloud and even Alexa couldn't access the skill code [1]. This makes the static code analysis of Alexa skills almost impossible. Although several dynamic testing tools have been developed in existing works, *e.g.*, SkillExplore [25], SkillDetective [43] and VITAS [31], they are rule-based on English and couldn't be applied to the non-English skills. This makes dynamic testing challenging because we need a method to understand skill outputs and generate responses in diverse languages. We aim to test all the skills in European marketplaces and get as many skill outputs as possible.

Given ChatGPT's excellent performance on NLP (Natural Language Processing) and sentence understanding, we used it for skill dynamic testing. In addition to understanding sentences and generating responses, another advantage of ChatGPT is its ability to interact with different languages. This eliminates the need to translate skill outputs to English during conversations and reduces potential errors. To evaluate the performance of ChatGPT on skill testing, we manually collected all the outputs from 100 skills and used the ChatGPT to interact with these skills. We used the "gpt-3.5-turbo" language model to get responses in our work. As a result, ChatGPT can effectively obtained 8 outputs while manual testing has 11 outputs on average. This shows that the ChatGPT can obtain the most outputs from a skill. Since existing tools are designed for English, we couldn't compare their performance with ChatGPT.

Alexa provides developers with a skill simulator to test developing skills. Meanwhile, it can also be used to invoke all skills in the skill store. In our testing, we used the skill simulator to invoke and test each skill in the European marketplace. To test the skills automatically, we used the Selenium Webdriver [13] to read skill outputs and fed ChatGPT responses in the simulator. Alexa requires each skill to provide three utterances and show them on the skill webpage for users to learn how to invoke a skill. For each skill, we obtained its invocation utterances from the skill store and inputted the utterances in sequence to invoke the skill. Note that one skill can be published to several marketplaces and the utterances can be in several languages. When a skill is invoked and provides an output in text, such as "What is your name?", we asked ChatGPT to give a brief response, such as an answer to a question. Then we fed the response obtained from ChatGPT to the skill simulator to get the following skill outputs. Such interactions would stop if no question or selection sentences are provided or it exceeds an iteration threshold we set (15 iterations).

## 6.2 Skills with Data Collection Through the Voice Channel

After one month of testing, we tested all the skills in 5 European marketplaces (GB, DE, ES, IT, FR). We gathered 247,797 voice channel outputs from 52,299 skills and translated them into English using Google Translate. To detect data collection behaviors in skill outputs, we used the same method in SkillExplore [25] and SkillDetective [43]. If any personal data semantically follows the word "your", *e.g.*, "your name", we consider it a data collection. We found 326 skills asking for different types of user data in European marketplaces. The most commonly asked data types are name, location, and email address.

After that, we checked the privacy policy inconsistency and GDPR non-compliance by comparing the data collection behaviors in skills with their privacy policies. For privacy policy inconsistency analysis, we detected the following three types of potential inconsistencies: If a skill collects user data but doesn't provide a privacy policy, we consider it as lacking a privacy policy. If a privacy policy is provided, but the collected data is not mentioned in the privacy policy, we consider it an incomplete privacy policy. If a skill collects user data through the conversational channel but claims it doesn't do that in its privacy policy, *e.g.*, "we don't collect any data from users", which is a deceptive privacy policy. We used PolicyLint [16] to find the negative statements in privacy policies. For GDPR non-compliance check, if a skill lacks a privacy policy, it violates GDPR compliance since it fails to provide all GDPR categories. For other cases, we need to check their GDPR compliance using the same method discussed in Section 5.

| Marketplace | Skills Collect Data | Lacking Privacy Policy | Incomplete Privacy Policy | Deceptive Privacy Policy | GDPR Violations |
|---|---|---|---|---|---|
| GB | 212 | 112 | 47 | 9 | 208 |
| DE | 74 | 33 | 20 | 0 | 67 |
| ES | 20 | 12 | 3 | 0 | 18 |
| IT | 9 | 5 | 2 | 1 | 8 |
| FR | 11 | 5 | 2 | 1 | 9 |
| Total | 326 | 167 | 74 | 11 | 310 |

**Table 5: Number of skills asking for user data through voice channel and skills having privacy policy inconsistencies.**

Table 5 shows the number of skills asking for user data through the voice channel and skills with privacy policy inconsistency in

each marketplace. After checking the privacy policy links of these skills, we found that 167 skills (51.2%) lack a privacy policy. For the other skills, 74 skills (22.7%) don't mention data collection in their privacy policies, indicating that only 26.1% of skills provide a complete privacy policy. Figure 5 in the Appendix presents a skill that asks for the user location but provides an unrelated page as its privacy policy without providing any useful information. We also found that 11 skills with data collection behaviors have a deceptive privacy policy stating they don't collect data. In addition, since these skills collect user data through the voice channel, they should follow the GDPR. However, we found that 310 skills (95.1%) have GDPR non-compliance issues in their privacy policies.

## 6.3 Skills with Data Collection Through Permission Requests

In addition to collecting user data through voice channel, Amazon Alexa also provides developers with different permissions [6] and some of them are about users' personal information. We consider a skill requesting such data collection permissions as another type of data collection behavior. In our work, we consider the following permission requests as sensitive data collection: First Name, Full Name, Email Address, Mobile Number, Device Address, Device Country and Postal Code, and Location Services. If a skill asks for any of the above permissions, it should provide a privacy policy that discloses the data collection.

| Marketplace | Skills Collect Data | Lacking Privacy Policy | Incomplete Privacy Policy | Deceptive Privacy Policy | GDPR violations |
|---|---|---|---|---|---|
| GB | 427 | 2 | 174 | 9 | 404 |
| DE | 206 | 0 | 70 | 0 | 190 |
| ES | 89 | 0 | 49 | 0 | 88 |
| IT | 74 | 0 | 22 | 2 | 73 |
| FR | 65 | 0 | 45 | 0 | 63 |
| Total | 861 | 2 | 360 | 11 | 818 |

**Table 6: Number of skills asking for data collection permissions and skills having privacy policy inconsistencies.**

Table 6 shows the skills that ask for data collection permissions and the number of skills with privacy policy inconsistencies in each marketplace. Surprisingly, two skills in the GB marketplace ask for data collection permissions without providing a privacy policy. However, developers need to submit permission and privacy policy information during the certification process and the Alexa platform can easily check such inconsistencies. Figure 6 in the Appendix shows one skill asking for permission but missing a privacy policy. For the other marketplaces, no skill lacks a privacy policy, possibly because they have removed such skills or have a more strict certification process. Unlike lacking a privacy policy, skills with an incomplete privacy policy exist in each marketplace (41.8% on average). We even found 11 skills providing a deceptive privacy policy. For example, the skill "Cursed Painting" asks for the user's "Email Address" permission, but it claims that "We do not collect any Personal Information". At last, we found that 95% of skills asking for permissions suffer from GDPR non-compliance issues. When comparing the two types of data collection behaviors, we found that the permission model can help improve the quality of privacy policies. Most skills asking for user data through voice

channel prefer not to provide a privacy policy. In contrast, skills requesting permission are less likely to do that. Nonetheless, *a significant number of skills collecting user data have GDPR non-compliance issues.*

## 7 DISCUSSION

### 7.1 Factors Causing GDPR Non-Compliance

After detecting many GDPR violations, we are interested in why such violations are prevalent in European marketplaces. We found general issues, *e.g.*, broken links, duplicate links, or unrelated pages. frequently appear in skills and lead to violations.

**Broken Links.** Skills with broken links couldn't provide any useful information in privacy policies, leading to possibly further violations. For a skill with a broken privacy policy link, if it has data collection behavior, it has an incomplete privacy policy since data collection is not mentioned. They also have GDPR violations in all categories since no GDPR category is provided. In addition, broken links will undoubtedly influence user experience.

**Duplicate Links.** There exists a large number of skills sharing the same privacy policy links and such an issue is mainly caused by the developers. Most developers, especially company accounts, prefer to use the same privacy policy for all their skills instead of writing a unique one for each skill. For example, the developer "Smart Skills" has 742 out of 992 published skills using the same privacy policy link and all these skills violate the GDPR requirements. Also, some developers might directly copy from other general privacy policies. To cover all the possible data practices, these privacy policies can be lengthy and users might skip reading them. For duplicate links, it is undoubtedly that most of them are inconsistent with skills' behaviors, leading to the same GDPR or privacy policy violations in skills. We found that 72% of skills using duplicate links in European marketplaces have GDPR violations.

**Privacy Policy Templates.** Similar to duplicate links, some skills use templates to generate privacy policies. Figure 7 in the Appendix shows the most commonly used template provided by "creator.voiceflow.com" and 2,636 skills in European marketplaces use the template. In this template, developers only need to change the app name and developer without modifying anything else. However, such a template claims the data collection without providing any information about GDPR, leading to all skills using the template in the European marketplace violating the GDPR requirements.

**Privacy Policies in English.** In addition, we found that 38% of the privacy policies in European marketplaces are written in English. This possibly indicates that the developers directly copied the privacy policy from the English marketplaces without considering the requirements of GDPR. The IT and ES marketplaces have the highest percentage of privacy policies in English, leading to them having more GDPR violations. Beyond GDPR violations, another potential issue is that non-English speakers may not comprehend English, rendering them unable to read the privacy policy.

**Unrelated Pages.** Developers might use an unrelated website, *i.e.*, the company website or advertisement page, as a skill's privacy policy. A skill that collects user data and uses such a privacy policy will violate GDPR compliance. For example, the "policy.com" website was used by several skills. However, it is an insurance company and developers might assume the website contains a privacy policy.

We discovered that in all marketplaces, there are skills using an unrelated page as their privacy policies. For example, a skill named "SwissGroove Web Radio" uses the company's main webpage as its privacy policy in all marketplaces, as shown in Figure 8 in the Appendix. Similar to the broken links, if a skill collects data but provides an unrelated page, it has an incomplete privacy policy and all types of GDPR violations.

## 7.2 Implication

Our work shed light on the current status of Alexa skills in European marketplaces regarding GDPR and privacy policy compliance. While most existing works analyze GDPR compliance for diverse targets [34–36], none analyzed the Alexa skills regarding GDPR compliance in privacy policies. Meanwhile, for Alexa skills, most researchers focused on privacy policy issues in the US or English-speaking marketplaces, but none of them worked on the skills in European marketplaces written in local languages. For the Alexa platform, it is important to be aware of the regulations for different places that it will target, *e.g.*, COPPA, HIPAA, CalOPPA, and GDPR. GDPR in Europe has stricter requirements than other areas and developers need to provide more information when writing a privacy policy. Although such regulation aims to protect users' data and privacy better, developers may overlook it. Our results show that the GDPR does have a positive influence on the privacy policies of European skills. European marketplaces under the GDPR, such as the GB, DE, and FR, perform better than other non-European marketplaces, such as ME and BR. Such results show the importance of GDPR and the necessity of setting up appropriate regulations in other areas, such as CalOPPA and CCPA.

## 7.3 Limitation

Our work has the following limitations. First, the performance of the trained model for detecting GDPR information can be improved by labeling a new dataset with more experts and sentences. Since existing datasets and privacy policies of skills may have different characteristics, labeling another skill privacy policy dataset may improve the model's performance. Second, more translation methods and tools can be tested to potentially improve the accuracy. In addition, we plan to use the ChatGPT (such as GPT-4), which performs well at language processing, to perform the translation. Third, the European skills might not be thoroughly tested because of the iteration limitation in our dynamic testing. Skills might hide more data collection behaviors in deep conversations, making them hard to find through dynamic testing. Even so, we tested all the skills in European marketplaces and identified hundreds of data collection behaviors in skill outputs. We plan to improve our tool and obtain more outputs from skills in our future work.

## 8 RELATED WORK

**Security and Privacy in Voice-apps:** More researchers and studies are drawing attention to the security and privacy of VPA platforms in recent years [15, 20, 21, 27, 29, 31, 38, 39]. Kumar *et al.* [28] discovered the squatting attack and analyzed the interpretation errors made by Amazon Alexa. They found that attackers can leverage systematic errors to direct users to malicious skills. Zhang *et al.* [44] found the voice masquerading attack, in which a malicious skill

impersonates the VPA service to steal a user's personal information. Cheng *et al.* [17] and Wang *et al.* [41] evaluated the certification system of VPA platforms and the results show that the skill vetting systems are untrustworthy. More recently, researchers applied dynamic testing or static analysis to find more privacy violations in skills. Liao *et al.* [32] checked the quality of privacy policies of Alexa skills and Google actions. SkillExplorer [25] tested 28,904 skills and found 1,141 skills requesting users' private information without providing a complete privacy policy. SkillDetective [43] tested 54,055 skills and found 6,079 policy violations, of which 623 skills were about data collection and privacy policy violations. Lentzsch *et al.* [30] found that 23.3% of privacy policies do not fully disclose the data types in requested permissions. SkillVet [19] evaluated the permissions system with privacy policies using a machine-learning based method and discovered 748 skills with an incomplete privacy policy. Xie *et al.* [42] tested over 20,000 skills and found 1,012 skills with privacy policy noncompliance issues. However, none of them tested and analyzed skills in non-English marketplaces.

**GDPR Non-Compliance Analysis:** The analysis of GDPR compliance in privacy policy started in 2018 following the enactment of GDPR and has attracted substantial attention in recent years. Tesfay *et al.* [40] labeled 45 privacy policies and proposed a machine-learning approach to analyze the privacy policy of GDPR. Gruschka *et al.* [24] discussed the state of legal regulations and analyzed the privacy-preserving techniques. Linden *et al.* [33] conducted a study comparing the privacy policies before and after enforcing GDPR. Fan *et al.* [22] checked the GDPR requirements and corresponding data practices in mobile health applications. Hamdani *et al.* [26] used NLP techniques to extract data practices from privacy policies and check their GDPR compliance. Liu *et al.* [35] annotated 36, 610 sentences to train deep learning models to analyze the compliance of privacy policies of Google Play apps. Rahat *et al.* [36] labeled 1,080 privacy policies of websites and trained a CNN model with active learning to classify sentences into 18 GDPR categories. Different from existing works, our work analyzed the privacy policies of Alexa skills in European marketplaces, which are inherently expected to adhere to the GDPR. We also compared the privacy policies in different marketplaces and languages, which indicates that GDPR has a positive influence on European privacy policies.

## 9 CONCLUSION

In this work, we conducted a comprehensive analysis of GDPR non-compliance within the privacy policies of Alexa skills in European marketplaces. We first gathered a skill privacy policy dataset about GDPR to trained a BERT model for classifying privacy policy sentences into GDPR categories. After analyzing all privacy policies of European skills, we found that GDPR noncompliance issues exist in 67% of European skills. We also designed a simple yet effective dynamic testing tool to explore actual data collection behaviors in European skills. After comparing the data collection behaviors with privacy policies, we found that 603 skills fail to provide a complete privacy policy and 1,128 skills have GDPR non-compliance issues. After comparing the violations in different marketplaces, we found the GDPR has a positive influence on the privacy policies in European marketplaces.

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

## Appendix A   GDPR NON-COMPLIANCE ISSUES OF EUROPEAN MARKETPLACES IN 2023

After our initial analysis, we recollected a skill dataset in 2023 and performed the same analysis to validate whether any changes appeared after one year. Table 7 shows the GDPR non-compliance issues in European marketplaces in 2023. Compared to 2022, the average violations per skill decreased in almost all European marketplaces while only the GB marketplace slightly increased, showing that more skills in European marketplaces are improving their privacy policies and following the GDPR better. Conversely, the average violation numbers in ME and BR marketplaces increased in 2023 without the limitation of GDPR, showing the necessity of regulation.

## Appendix B   OFFICIAL SKILLS

Besides the third-party developers, Amazon also publishes skills on the skill store, which we call "official skills". We find several developer accounts potentially belonging to Amazon, such as "Amazon" and "Amazon Prime Video", in European marketplaces, For official skills, there is a higher expectation for them to follow the requirements better and provide good examples for other developers. However, for the 55 Amazon official skills in European marketplaces, almost all of them use an Amazon privacy notice in the US marketplace as their privacy policy instead of the privacy policy versions in European languages. Also, Amazon's privacy policy doesn't provide enough information about the GDPR, which leads to 54 skills violating GDPR compliance. Interestingly, one official skill in the DE marketplace uses a privacy policy template instead of Amazon's privacy policy. This skill also has GDPR violations. The violations in Alexa official skills show their overlook of the GDPR and privacy policy quality.

## Appendix C   EXAMPLES OF SKILL PRIVACY POLICIES

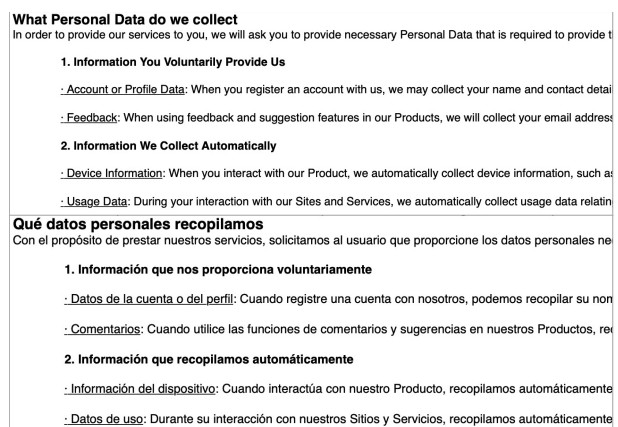

**Figure 2: The privacy policies of skill "WOOX Security" in English and Spanish. The content, titles, and headings in the two privacy policies are aligned.**

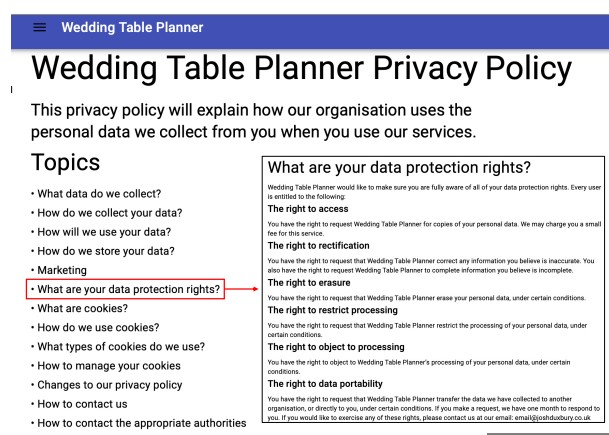

**Figure 3: A skill privacy policy that strictly complies with all GDPR requirements.**

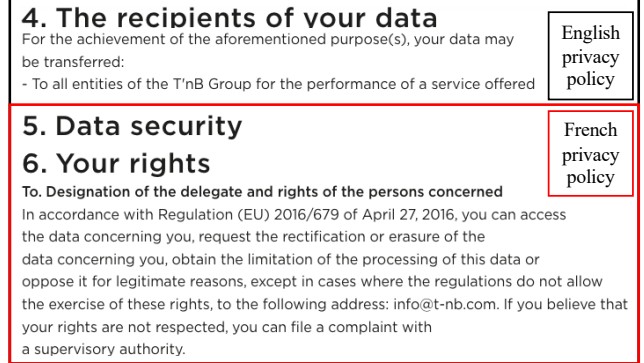

**Figure 4: One privacy policy adds new content in the European marketplace to comply with GDPR. The French privacy policy has already been translated into English, and the red box content is newly added for GDPR compliance compared to the English version.**

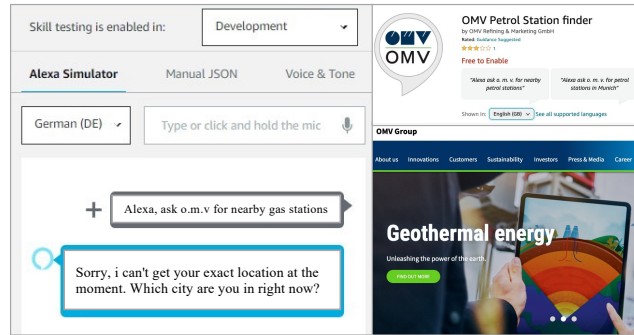

**Figure 5: A skill asks for user location through voice channel but provides an incomplete privacy policy.**

| | European Marketplaces | | | | | Comparison | | | |
|---|---|---|---|---|---|---|---|---|---|
| GDPR category | GB | DE | ES | IT | FR | US | ME | BR | Average |
| Data Retention Period | 41.06% | 21.31% | 34.82% | 39.18% | 35.07% | 29.07% | 37.02% | 54.70% | 36.53% |
| Data Processing Purposes | 3.68% | 2.65% | 3.57% | 3.91% | 3.97% | 4.71% | 3.30% | 16.69% | 5.31% |
| Contact Details | 4.30% | 4.87% | 7.00% | 6.04% | 5.96% | 5.79% | 8.25% | 22.59% | 8.10% |
| Right to Access | 32.63% | 49.08% | 40.76% | 36.10% | 41.20% | 40.20% | 44.69% | 48.98% | 41.71% |
| Right to Rectify or Erase | 39.03% | 22.12% | 32.57% | 43.13% | 31.27% | 24.21% | 30.56% | 53.36% | 34.53% |
| Right to Restrict of Processing | 26.70% | 22.72% | 31.34% | 31.16% | 33.72% | 33.47% | 33.00% | 43.40% | 31.94% |
| Right to Object to Processing | 50.85% | 28.85% | 50.44% | 51.30% | 46.29% | 42.52% | 51.15% | 67.89% | 48.66% |
| Right to Data Portability | 56.66% | 35.18% | 51.80% | 56.00% | 51.08% | 47.54% | 56.89% | 68.46% | 52.95% |
| Right to Lodge a Complaint | 60.11% | 37.96% | 60.21% | 55.77% | 57.86% | 55.25% | 66.43% | 78.87% | 59.06% |
| # of Skill with Data Collection | 6041 | 3061 | 1627 | 1839 | 1279 | 1335 | 1049 | 1327 | |
| # of Skill with Violations | 5680 | 2654 | 1572 | 1749 | 1191 | 1256 | 1005 | 1314 | |
| Percentage of Skill with Violations | 94.02% | 86.70% | 96.62% | 95.11% | 93.12% | 94.08% | 95.81% | 99.02% | |
| Average Violations per Skill | 3.87 | 3.02 | 3.84 | 3.95 | 3.81 | 3.52 | 4.07 | 5.39 | 3.70 |

Table 7: GDPR violations of all marketplaces in 2023.

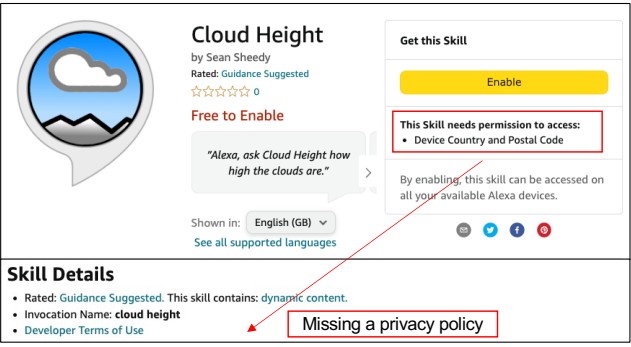

Figure 6: A skill asks for data collection permission but lacks a privacy policy.

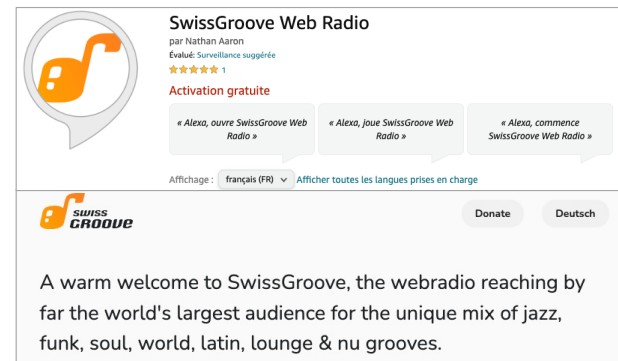

Figure 8: A skill uses an unrelated page as its privacy policy in all marketplaces.

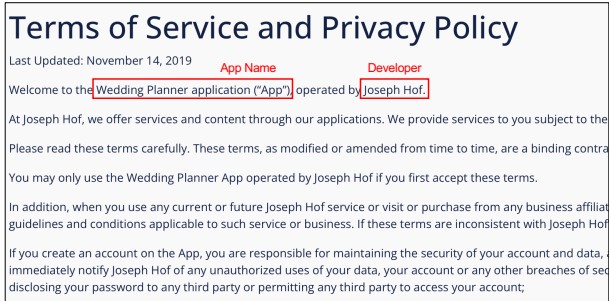

Figure 7: The most commonly used template provided by "creator.voiceflow.com" and 2,636 skills in European marketplaces are using the template.

