# OpenReview forum: "Understanding GDPR Non-Compliance in Privacy Policies of Alexa Skills in European Marketplaces"
_ACM.org/TheWebConf/2024/Conference — TheWebConf24_

### Official Review · Reviewer_K1S4 · 2023-11-21

**Novelty:** 3
**Technical Quality:** 5

**Review:**

This paper analyzes GDPR violations in the Alexa Skill marketplace. To study GDPR violations, they first collected the privacy policies of all Alexa Skills. Then, they translated them into English and predicted which sentences were relevant to GDPR compliance. Then they manually coded violations of the policy by searching for whether every category required by the GDPR is represented in the privacy policy. They provide descriptive statistics about the violations found (like relative numbers of violations across different countries’ markets). They also used ChatGPT to try to determine if Skills are asking for personal information without having given appropriate notice in the privacy policy.

The overall finding that a large majority of Alexa skills have GDPR violations is very interesting and would have significant policy implications if validated.

A few questions and comments on the methodological approach:

- Why is the sentence-level classification task necessary if you eventually do a keyword search for each GDPR category anyway?
- I would have liked to see more description of the sentence classification task. What are the labels in the datasets used to train the model? Does each dataset have the same set of labels? If not, how did you reconcile labels across datasets?
- I would have liked to see a quantitative comparison between the performance of your sentence classifier and others you trained or those proposed in the literature. Did you try fine-tuning models? This can yield better results, especially given limited computation, than training from scratch.
- I would have also liked to have seen more analysis of the performance of the model and what biases this may introduce into the subsequent GDPR violation analysis.  How might the predicted GDPR-relevant differ from the ground truth? Will this introduce bias? Without more details on these questions, it is difficult to evaluate the subsequent analysis of GDPR violations. I also have concerns about the dynamic skills testing. The paper says: “If any personal data semantically follows the word “your”, e.g., “your name”, we consider it a data collection.” but I imagine that many non-data collection-related sentences might use “your”. How do we validate that these are actual instances of data collection versus a false positive like “What is your answer?” in an Alexa game of Jeopardy.

**Questions:**

See the bullets above.

**Reviewer Confidence:**

3: The reviewer is confident but not certain that the evaluation is correct

**Scope:**

2: The connection to the Web is incidental, e.g., use of Web data or API

---

### Official Review · Reviewer_CJNj · 2023-11-22

**Novelty:** 6
**Technical Quality:** 6

**Review:**

The paper presents an analysis of the General Data Protection Regulation (GDPR) compliance in the context of Alexa skills' privacy policies within European marketplaces. The methodology includes data collection, classification model development using BERT, and dynamic testing using ChatGPT. The paper is well-organized with clear sections, including background, methodology, analysis, and discussion. Concepts are explained in detail, aiding comprehension for readers who may not be familiar with GDPR or Alexa’s skill ecosystem.

The focus on Alexa skills in the European marketplaces regarding GDPR compliance is a relatively unexplored area, making this work original in its scope. With increasing concerns about data privacy and the widespread use of voice assistants, this research is pertinent. The findings may provide insights for developers, policymakers, and researchers in understanding and improving GDPR compliance in VPA platforms.

Pros

Covers a wide range of marketplaces and skills, providing a holistic view of GDPR compliance in this context.

Offers actionable insights for improving privacy policy compliance in voice assistant platforms.

Employs robust data collection and analysis methods, enhancing the study's reliability.

Cons

The technical details of the methodology might be challenging for readers without a background in AI or data analysis.

The study is focused on European marketplaces, which might limit its applicability to other regions.

Given the rapidly evolving nature of technology and regulations, some findings might become outdated quickly.

**Questions:**

How adaptable is your methodology to account for changes in GDPR regulations or updates in Alexa's skills' functionalities?

Could your findings and insights be generalized to other voice assistant platforms beyond Alexa, such as Google Assistant or Apple's Siri?

How do cultural and legal differences across European countries impact GDPR compliance in Alexa skills' privacy policies?

Could you elaborate on any limitations you encountered using BERT and ChatGPT in your analysis, and how these might have impacted your results?

Did your research uncover any notable variations in GDPR compliance across different European marketplaces, and if so, what might be driving these differences?

**Ethics Review Description:**

The research involves analyzing privacy policies and potentially user data or interactions with Alexa skills. It is crucial to ensure that the data used in the study was obtained and processed in a manner that respects user privacy and adheres to consent requirements.

**Ethics Review Flag:**

Yes

**Reviewer Confidence:**

3: The reviewer is confident but not certain that the evaluation is correct

**Scope:**

3: The work is somewhat relevant to the Web and to the track, and is of narrow interest to a sub-community

---

### Official Review · Reviewer_juzK · 2023-11-22

**Novelty:** 5
**Technical Quality:** 4

**Review:**

In this research authors have analyzed privacy policies of European voice personal assistants regarding GDPR. The study presents interesting conclusions about the services provided, indicating that most of them do not follow GDPR. The automatic process described to reach the conclusions is useful for the monitoring of such services in the future and opens space to improvements in the automatic methodology.  Regarding the results, it would be good to see more effort to assert the quality of the automatic method presented. Authors could, for instance, take samples of the classification produced and ask specialists to classify the services in such samples, showing at the end the accuracy of their method.

**Questions:**

Could you give more information about the quality of the method presented, including more validation with specialists to check, for instance, whether a sample of services would receive from the specialists the same classification assigned by your method ?

**Reviewer Confidence:**

3: The reviewer is confident but not certain that the evaluation is correct

**Scope:**

4: The work is relevant to the Web and to the track, and is of broad interest to the community

---

### Official Review · Reviewer_Zy19 · 2023-11-24

**Novelty:** 6
**Technical Quality:** 6

**Review:**

***Paper summary:***

The study examines the GDPR compliance of voice apps, known as skills, in European marketplaces. The research focuses on privacy policies and data collection behaviors of these skills. Using a large dataset and a BERT model for classification, the analysis reveals that a significant portion (67%) of privacy policies fail to comply with GDPR. Among skills with data collection, half lack complete privacy policies, and 95% exhibit GDPR non-compliance issues. The study notes a positive impact of GDPR on European privacy policies compared to non-European marketplaces.




***Detailed comments for authors ***

Thank you for submitting your paper on GDPR Non-Compliance in Privacy Policies of Alexa Skills in European Marketplaces. I find the topic quite interesting, and I appreciate the effort you have put into your research. Below, I provide a detailed review:

**Reasons to Accept the Paper:**

- Well-Written and Organized:
        The paper is nicely written and well-organized, making it easy to follow.


- Robust Data Analysis:
        The use of a large dataset consisting of 23,927 privacy policies is commendable. Moreover, training a BERT model for predicting GDPR-related sentences and conducting a large-scale analysis on GDPR non-compliance demonstrates thorough research.


- Open Source Dataset:
        The decision to make the dataset open source adds value to the research community.


- In-Depth Analysis in Section 6:
        Section 6, where you analyze the inconsistency in privacy policies by comparing them against actual data collection behaviors, is particularly insightful.


**Reason Not to Accept the Paper:**

While analyzing voice apps is relatively new, the examination of privacy policies and GDPR non-compliance has been extensively studied.
Also, the conclusion that GDPR has a positive influence on European privacy policies has already been validated by previous works. I suggest extending the related work section to discuss in more detail how your work compares to previous studies analyzing privacy policies.

**Questions:**

- Can you provide more context on the existing studies that have analyzed privacy policies and GDPR compliance, and how your work builds upon or differs from these previous contributions?

**Reviewer Confidence:**

3: The reviewer is confident but not certain that the evaluation is correct

**Scope:**

4: The work is relevant to the Web and to the track, and is of broad interest to the community

---

### Official Review · Reviewer_7nNZ · 2023-11-25

**Novelty:** 5
**Technical Quality:** 5

**Review:**

**Summary**

- This paper delves into privacy concerns surrounding third-party developed applications, or ‘skills’, on Amazon’s Alexa platform.
- The focus is on European skills compliance with GDPR, where stringent data collection and processing rules apply.
- A large-scale, European skill dataset is analysed, and BERT and ChatGPT based testing tools are used to check the conformity of skill privacy policies with GDPR and their consistency with actual data collection behaviours.

**Strong points**
- The topic is relevant and timely in the current age of data privacy concerns and GDPR implications upon data collection practices.
- The use of GDPR as a reference point ensures a clear and standardised benchmark for analysis.
- The methodology, involving a combination of policy analysis and testing tool application, is interesting.
- The paper contributes to the community by sharing the dataset, model, and results.

**Weak points**

- The authors have invested significant effort to understand the GDPR compliance practices for skills in Alexa Skills Store. Yet, the document fails to clarify why this issue bears importance and how the authors’ findings may apply to other websites or applications.
- The description of the training and validation dataset for the BERT and translation methodologies could be improved. It is unclear if the authors have used the 2586 sentences to train the BERT model, or whether these are only for validation purposes. Besides, it would help to know if the translation evaluation dataset is balanced. The Macro F1 score might also prove crucial, specifically if the dataset is imbalanced.
- The authors’ ChatGPT based method to simulate the data collection process followed by skills is indeed interesting. Although the paper claims that any personal data following the word “your” is considered data collection, the accuracy of this method is unclear. For instance, is “your questions” also categorized as data collection?

**Questions:**

see weak points above

**Reviewer Confidence:**

3: The reviewer is confident but not certain that the evaluation is correct

**Scope:**

3: The work is somewhat relevant to the Web and to the track, and is of narrow interest to a sub-community

---

### Decision · Program_Chairs · 2024-01-22

**Decision:**

Accept

**Comment:**

Our decision is to accept. Please see the AC's review below and improve the work considering that and the reviewers' feedback for cemera-ready submission.

"The paper analyzes the privacy policies of Alexa skills and assesses whether they comply with the GDPR requirements. The reviewers appreciated the comprehensiveness of the analysis and the fact that the datasets have been open-sourced. Additionally, the reviewers found the adapted methodology to use ChatGPT to simulate the data collection behaviors of skills to be interesting."